# A Study on the Design of Vision Protection Products Based on Children’s Visual Fatigue under Online Learning Scenarios

**DOI:** 10.3390/healthcare10040621

**Published:** 2022-03-25

**Authors:** Di Feng, Chunfu Lu, Qingli Cai, Jun Lu

**Affiliations:** 1Department of Industrial Design, School of Design and Architecture, Zhejiang University of Technology, Hangzhou 310023, China; fengdi@zjut.edu.cn (D.F.); 2111915007@zjut.edu.cn (Q.C.); 2111915013@zjut.edu.cn (J.L.); 2Institute of Industrial Design, Zhejiang University of Technology, Hangzhou 310023, China

**Keywords:** children’s visual fatigue, online learning, screen view parameters, vision protection product

## Abstract

The rate of myopia in children is increasing rapidly under online learning scenarios. One of the important reasons for this is incorrect reading and writing posture. Three screen view parameters (viewing angle, viewing height, and viewing distance) are selected as significant influencing factors and blink rating is used as a sign of visual fatigue through literature analysis to study the influence factors of myopia in children, and their correlation. Children’s visual fatigue is evaluated by subjective evaluation and is recording using an eye tracker for changes in the three factors through online learning scenario simulation experiment. An optimal regression model is constructed that illustrates the relationship between the three variables and the visual fatigue levels. The aim of this study is to confirm the quantitative relationship between the screen view parameters and visual fatigue, and to design a child vision protection product on this basis. The test results show there is a linear positive correlation between the viewing angle, viewing height, and viewing distance. A vision protection device has been designed based on this model and was verified through function prototype testing. The result of this study quantified the relationship among screen view parameters and children’s visual fatigue, which provides a theoretical basis for the design of a children’s visual protection device.

## 1. Introduction

With the development of network technology and the popularization of various novel electronic products, the myopia rate of children continues to increase [1,2,3]. Since the COVID-19 outbreak, the slogan “classes suspended but learning continues” has accelerated the development of online education, increasing the frequency of screen use among children. Existing studies have suggested that the prevalence of myopia in grade 3 children increased from 13.3% at the end of 2019 to 20.8% at the end of 2020 [4]. In addition, wrong eye posture and prolonged irregular eye use are also some of the important factors leading to vision loss [5,6,7]. Most studies in China and abroad regarding the impact of reading and writing postures on myopia have focused on examining the viewing distance, chest–table distance, viewing angle, and viewing duration. The correlation of reading and writing postures with myopia is still unclear and requires further studies. 

Visual fatigue is usually manifested by a series of eye discomforts and symptoms, including fatigue, dryness, pain, burning sensation, redness, difficulty concentrating, and double vision [8]. In severe cases, visual fatigue can progress to headache, dizziness, fatigue, and other general discomfort. The degree of visual fatigue also increases with aging [9]. However, existing local and international studies mostly focus on investigating visual fatigue in adults, with relatively less studies in children. 

The occurrence of visual fatigue is closely associated with reading and writing postures [10], display type [11,12], screen parameters [13], and the environment [14,15]. Previous studies have pointed out that when the preferred viewing distance and vertical gaze inclination are 90 cm and −10°, respectively, there will be less visual fatigue when viewing visual displays [16]. When viewing different types of electronic screens, the effects of viewing distance and viewing time on visual fatigue are different [17], in addition, the preferred viewing distance and preferred viewing angle of the electronic paper display are 495 mm and 30–35° [18]. In previous studies of visual fatigue, one widely examined screen parameter is viewing distance, followed by viewing angle and viewing height [19]. However, existing studies on the correlation between visual display terminal and visual fatigue have barely examined all three parameters simultaneously. 

Visual fatigue can reflect the degree of visual fatigue of the subjects through subjective evaluation and objective measurement. Objective measurements obtain the relevant physiological parameters of the subjects based on biological signals such as electroencephalography (EEG) [20], electrocardiography (ECG) [21], electrooculography (EOG) [22], blink frequency (BF), critical flicker frequency (CFF), child diameter, and other eye movement indicators. Among them, the testing process of biological signals is complex and requires subjects to contact the probe of the measuring instrument. The eye movement indicators are simple to obtain and make it easy to evaluate the subjects’ visual fatigue. Many researchers have suggested a correlation between BF and visual fatigue, where the BF increased over time, but the rate of the increase showed a downward trend and slowed down after 30 min [23,24]. Based on the test conditions and effectiveness of the indicators, this study determined BF as an eye movement indicator for objective evaluation, and combined subjective evaluation to obtain the measurement of visual fatigue.

This paper proposed a model to demonstrate the correlation between screen view parameters (viewing distance, viewing angle, and viewing height) and BF, and validated the model by designing a flexible reading table and assessing the model fit. This study is divided into two stages, as follows: (1) the subject was invited to be passively engaged in the human–machine experiment (i.e., video watching). Then, the subjective assessment scale for visual fatigue was used to obtain the data on screen parameters from each subject and the BF data were also collected. Based on the collected data, a linear regression model was constructed to model the relationship between BF and screen view parameters, and (2) the overall design of a vision protection device was proposed based on the abovementioned linear regression model (including product appearance, structure and hardware program), and product testing was performed to validate the regression model and product effectiveness. This study aimed to offer data references for the design of vision protection products to protect eyesight during screen time and to develop market products based on theoretical models.

## 2. Study Methods

### 2.1. Participants

All participants had normal or corrected-to-normal vision; no abnormality in binocular visual function; no functional dyslexia; and no eye diseases, such as strabismus, amblyopia, or astigmatism. Before the start of the study, the study purpose was explained in detail to all subjects, and the informed consent form was obtained from all participants. To avoid interfering with the assessment of visual fatigue, subjects were advised to have adequate sleep the night prior to the study, as well as proper use of their eyes, and no reading or no use of electronic devices were allowed within 1 h before the onset of the study. 

### 2.2. Parameters Calculation

This study evaluated three independent variables: viewing distance (viewing distance L), referring to the straight-line distance between the eyes and the center of the screen; viewing height (viewing height H), measured vertically from the horizontal line of sight and the tabletop to the center of the screen; and viewing angle (viewing angle α1 as shown in Figure 1), measured as the angle between the device and the horizontal tabletop. At the same time, the angle between the horizontal line of sight and the screen was recorded as α2. Clockwise, an upward-forming viewing angle represented a positive measure, whereas a downward-forming viewing angle represented a negative measure. Forward head posture and neck bending can cause postural fatigue [25,26]. The subjects were asked to lay their hands crossed on the table; keep their legs together without swinging; sit up straight without bending forward; and keep their shoulders, neck, and waist in a straight line. This test aimed to examine the correlation of visual fatigue with the three independent variables. To enable changes in screen height, screen angle, and screen distance, we prepared a simplified test stand for the test to simulate flexible iPad workstations. Two rotating shafts and control levers were installed below the working stand to enable screen angle and height changes.

Based on Heue H. [27] and Rafael I. [28], the study designed a more appropriate subjective visual fatigue evaluation scale, including ocular symptoms, headache symptoms, and body pain, to better distinguish different degrees of fatigue symptoms. In the experiment, the subjective evaluation consisted of a pre-test and a post-test, and each item was rated on a five-point scale.

### 2.3. Study Setting

The study lasted from 19 September 2021 to 15 January 2022 at the man–machine laboratory of Zhejiang University of Technology, which is approved by Institute of Industrial Design of Zhejiang University of Technology. An iPad Pro 11 produced by Apple, and manufactured in Shenzhen, China (screen size, 12.9 inches; screen resolution, 2732 × 2048; display brightness, maximum (474 nit)) was used as the screen display in this study. A phone was used to film and record the subjects during the study. Other materials included an SMI head-mounted eye tracker; a simplified stand able to adjust viewing distance, angle, and height; a set of common desk and chair for primary school students; an ultrasonic distance measure; a measuring tape; and an angle gauge.

To avoid interference with the study results, any factor that may have affected the study results was controlled prior to the study. As for the task design, the subjects were asked to watch “Life’s Little Secrets”, a 30-min video from a primary school popular science series, and complete a simple test paper at the end of the video to determine their reading efficiency. The study was conducted in a room that could control different types of glare, with the windows covered with curtains. The lights that illuminated the room were not placed above the screen. Ambient lighting was controlled at 500 lx and the ambient temperature was maintained at 26 °C. The subjects needed to sit on the prepared chair and after adjusting the distance between the desk and chair, both objects were remained in place. During the study, the subjects were not able to change their postures at will. To prevent the subjects from having severe ocular discomfort, the study would be terminated in case of any obvious symptoms of ocular discomfort, such as tear secretion by the lacrimal ducts due to eye pain, bloodshot eyes, or lethargy, as shown in Figure 2.

### 2.4. Study Process

Before the onset of the study, the basic data of the subjects, including sitting height, eye height, and corrected-to-normal vision, were recorded. The subjects were then asked to sit in the reference posture, and then adjust the angle, height, and distance of the iPad screen based on their viewing habits. The experimenter measured and recorded the data, and reminded the subjects that they could reduce postural strain by slightly twisting their body. The shooting and recording equipment were placed right in front of the subjects, so that the eye blinks of the subjects could be properly recorded. The subjects were also asked to fill in the subjective assessment scale for visual fatigue. We then performed a five-point calibration and validation procedure using the SMI head-mounted eye tracker on the subjects. After reliable tracking of the eye movements was guaranteed, the subjects were asked to watch a 5-min scenery video (serving as the blank control group), followed by “Life’s Little Secrets”, during which the subjects’ physical status was observed, including the eyes, limbs, and body. Afterwards, the subjects took the test regarding video content and filled in the subjective assessment scale for visual fatigue. During the study, the subjects did not have vision or hearing impairments, nor were they exposed to physical or chemical stimuli that could cause a blink reflex. The experimental flow chart is shown in Figure 3.

### 2.5. Statistical Analyses

To explore the relationship between screen viewing time and subjective fatigue evaluation, the paired t-test was performed on the subjective visual fatigue evaluation of the subjects before and after the test. The number of eye blinks measured by the eye tracker every 5 min was taken as the experimental group, and the number of eye blinks while watching the scenery video was taken as the blank control group. In order to explore the relationship between screen viewing time and visual fatigue, statistical analysis was carried out on the blink frequency data of each time period and the blank control group. 

After obtaining the data of three screen view parameters and blink frequency through an online learning scenarios simulation experiment, a multivariate linear regression model was used to reveal the linear relationship between the explained variable and other variables. The mathematical model is as follows: (1)$$y=\beta_{0}+\beta_{1}\chi_{1}+\beta_{2}\chi_{2}+\cdots +\beta_{i}\chi_{i}+\varepsilon$$
where β_0_ is the regression constant, β_0_, $\beta_{1}$, β_2_⋯ are the undetermined parameters, and *ε* is the constant. 

The vision protection device designed based on the formula model that incorporates the three important factors for visual fatigue (viewing angle, viewing distance, and viewing height) was finally used for testing and verification. In the validation experiment, the post-test results of the subjective visual fatigue evaluation of the formal test and the verification experiment were compared and analyzed.

## 3. Results

### 3.1. Data Processing and Analysis

In total, 52 grade 3–4 children (26 boys and 26 girls; mean age, 9.4 ± 0.6 years) participated in this study. The paired t-test results showed that the subjective fatigue evaluation after the test was significantly higher than that before the test, and there were extremely significant differences in the subjective fatigue scores of the subjects in terms of eye symptoms, headache symptoms, and body pain (*p* < 0.01), as shown in Table 1. As can be observed during the study, the children began to yawn and rub their eyes after watching the video for 15 min, and at 20 min, they started to accommodate visual fatigue by prolonged eyelid closure and rapid blinking, suggesting a certain degree of fatigue being present. 

The paired sample t-test results of the blank control group and the experimental group are shown in Table 2. Periods of 0–30 min were intercepted for display because it has been found that BF tends to be stable after 30 min when children watch on an electronic screen. The blinking frequency first increased and then decreased, and reached the maximum value (M ± SD = 10.680 ± 3.074) at 25 min. There was no statistical difference in the BF in the first 10 min of watching the screen (*p* > 0.05), but there was a statistical difference in the change in the frequency when the video viewing time was 15 min (*p* < 0.05). There were significant differences in the changes of BF when the video viewing times were 20 min, 25 min, and 30 min (*p* < 0.01). There was a significant positive correlation between the screen viewing time and BF, and visual fatigue began to appear when the screen viewing time was 15 min.

The data statistics of the preferred viewing angle, apparent height, viewing distance, and the angle between the line of sight and the screen are as follows: the preferred viewing angle of the subjects was 21.86 ± 2.22°, the preferred mean viewing height was 198.56 ± 12.08 mm, the preferred viewing distance was 531.26 ± 19.89 mm, and the angle between the line of sight and the screen was 81.87 ± 3.39°.

Stepwise regression analysis was performed to include or exclude a stepwise particular independent variable from all of the optional independent variables, until the optimal regression equation was established, as shown in Table 3.

As can be seen from Table 4, Equation (2) was the optimal model and the viewing angle had the biggest impact on BF with a standardized regression coefficient of 0.507, followed by viewing distance (0.290) and viewing height (0.221). 

The optimal linear regression equation was then constructed with BF(y) as the dependent variable and the viewing angle (*X*_1_), viewing distance (*X*_2_), and viewing height (*X*_3_) as the independent variables, as follows: (2)$$y=0.507X_{1}+0.290X_{2}+0.221X_{3}-26.149$$

The determination coefficient of the equation was *R*^2^ = 0.948. The regression coefficients suggested that for every 1° increase in the viewing angle (*X*_1_), BF increased by 0.507 times/min; similarly, for every 1 mm increase in the viewing distance (*X*_2_) and every 1-mm increase in the viewing height (*X*_3_), BF increased by 0.290 and 0.221 times/min, respectively.

### 3.2. The Design and Validation of Vision Protection Device

Given the difficulty in implementing function and production costs, the viewing height was determined by the input of the user height and the desktop height so as to reduce the calculations. Only the constantly changing viewing angles and viewing distances during product usage needed to be considered in the product design logic.

The size of the arm-protecting clamp that could support electronic products of different sizes (phones and pads) was designed based on the online learning environment. A gearbox with endless screws was designed according to the linear relationship between the viewing angle and the viewing distance, so that linear adjustment of the viewing distance and the viewing angle could be realized by swinging the rocker arm, as shown in Figure 4.

An ultrasonic ranging sensor was used to measure the distance from the electronic product to the face (considered as the distance between the electronic product and the eyes). The motor rotated and drove the gearbox to adjust the angle of the arm-protecting clamp. A potentiometer was used to conduct precise control of the rotation angle.

After the electronic product–face distance generated by the ultrasonic ranging sensor was entered, the rotation angle was produced and fed into the motor. The designed programs included battery control, ultrasonic ranging (for measurement of viewing distance), motor control (to drive the rotation of the gearbox), and potentiometer-assisted angle control (for precise control of the angle), of which the motor control program adopted the linear regression equation. The angle of the rocker arm of the vision protection device could be automatically adjusted to the optimal value according to the sight distance. The core control program is as follows:

void csb_trig_on(void) // 10 * 100 ms

{

if(gMode.status == STATUS_CLOSE || gMode.status == STATUS_PLACE) { rf_cnt = 0; return; }



rf_cnt++; 

if(rf_cnt >= 5) {

rf_cnt = 0;

d2 = d1;

d1 = distance1;

gDistance = (d1 + d2)/2; 

Distance_Trig();

y = 0.507*angle + 0.290*distance1 + 0.221*h-26.149;

return y;

}

The verification results are shown in the Figure 5. Subjective eye fatigue assessment mainly included three aspects of eye symptoms, head symptoms, and physical symptoms. Each question was divided into five grades, from no to strong feeling, corresponding to 0–4 points, respectively, with a total of 10 questions and a total of 40 points. The results showed that the mean values of the ocular symptoms, headache symptoms, and body pain of the subjects in the validation experiment were all lower than those in the formal experiment, and the reduction in ocular symptoms was the largest. 

## 4. Discussions and Conclusions

This study aimed to investigate the effects of the viewing angle, viewing height, and viewing distance on visual fatigue when viewing electronic screens. Through experiments, it was found that there was a linear regression effect between human–machine viewing parameters and blink frequency, and the viewing angle, viewing height, and viewing the effects of distance on visual fatigue were all positive. In the formal experiment, the subjects began to experience visual fatigue from 15 min, which showed that the subjects had a significant perception of subjective visual fatigue (*p* < 0.05) and there was a significant difference in the BF from 15 min (*p* < 0.05). The BF of the patients showed an increasing trend with the prolongation of time and showed a decreasing trend after 25 min, and the blink frequency was the largest at 25 min (M ± SD = 10.680 ± 3.074). Periods of 0–30 min were displayed because the data were basically flat after 30 min. This result was basically consistent with the research findings of Zhang Y. et al. [11]. In the study of Zhang Y. et al., BF showed a downward trend after 30 min. A possible explanation for this is that the object of this study was children, and the visual fatigue performance of children is different from that of adults. Eyes also have different adaptations to screens [29], thus showing that children experience visual fatigue earlier than adults. In the verification experiment, the subjects began to experience visual fatigue from 20 min, mainly manifested as a significant difference in BF from 20 min (*p* < 0.05). After 25 min, there was a downward trend, and at 25 min, the largest blink frequency (M ± SD = 10.788 ± 3.261) was found, but the formal experiment started to show visual fatigue at 15 min, which was 5 min earlier than the verification experiment. This result shows that the model was effective at alleviating visual fatigue. At the same time, according to the subjective visual fatigue evaluation, it can be seen that there were extremely significant differences between pre- and post-test in terms of eye symptoms, headache symptoms, and body pain (*p* < 0.01). It can be seen from the interviews of the subjects that after watching a video for a period of time, children felt that their eyes were sore, and they wanted to look away to see other places. They believed that looking into the distance would have a certain effect on alleviating eye fatigue, and according to the main test observation, the children began to yawn and rub their eyes after 15 min to relieve eye fatigue [30]. 

The study found that there are different degrees of correlation between different human–machine viewing parameters and blinking frequency [31,32], among which the angle of view and blinking frequency were extremely significantly positively correlated (*p* = 0.951), and the visual height was significantly positively correlated with blinking frequency (*p* = 0.915). There was a significant positive correlation between the visual distance and the blink frequency (*p* = 0.906), indicating that within a certain range, the blink frequency would increase with the increase of the angle of view, and would also increase with the increase in the visual distance and the visual height. The regression coefficient shows that the angle of view had the greatest influence on the blink frequency (0.507), followed by the viewing distance and the apparent height. Therefore, when studying the visual fatigue caused by electronic screens, the viewing angle, viewing distance, and visual height should be taken into account [33,34]. It can be seen from the verification experiment of the product that the visual fatigue of the subjects was delayed compared with the formal experiment, indicating that the construction of the regression model was effective.

In this paper, the regression equation model constructed by the viewing angle, visual distance, visual height, and BF was applied to the functional design and development of the vision protection device, and the verification experiment of the vision protection device was carried out. The results show that the use of the vision protection device can alleviate visual fatigue. Therefore, after the vision protection device solution was determined, 50 small-batch trial-produced products will be put into Zhejiang Province, China for regional inspection. After confirming that the products have no obvious problems, they will be put into the Chinese market for extensive verification by a wide range of users. Based on this method, a series of children’s visual protection products can be developed to control more influence factors of visual fatigue, and provide human–machine data, and innovative application reference for the design of auxiliary electronic equipment viewing products.

Because of the limitations of the experimental equipment and other force majeure factors, there are still some limitations in the experimental results of this paper: (1) The age of the subjects in this study was around 9 years old and hence the study results might not apply to older or younger children. In the follow-up study, the age will be broadened for experimental research and the scope of application of the model will be increased. (2) The study was conducted on subjects from Zhejiang province, and children in other regions might have different physiological conditions. In the future, a more accurate regression equation will be obtained based on the big data feedback after the product is put on the market, so as to improve the product program. (3) Subjects were asked to consciously constrain their posture during the study, which might not be sufficient to ensure stability of viewing angle, viewing height, and viewing distance. Subsequent research will consider mechanical fixation of the subjects’ posture when watching electronic screens. (4) The electronic screen used was an iPad screen, the study results might not apply to bigger screens (tablet PC) or smaller screens (phone). (5) This experiment only studied three screen view parameters of children’s visual fatigue under online learning scenarios because of limited laboratory equipment and time. Comparative experiments have not been recommended to determine the dominant influence factor.

## Figures and Tables

**Figure 1 healthcare-10-00621-f001:**
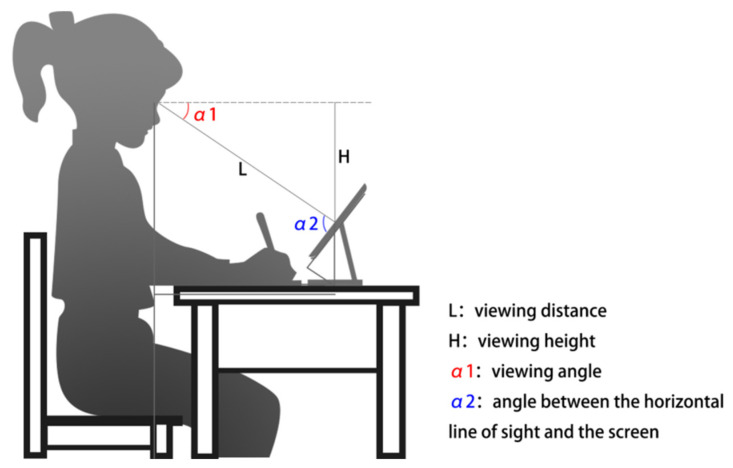
Independent variable measurement definition.

**Figure 2 healthcare-10-00621-f002:**
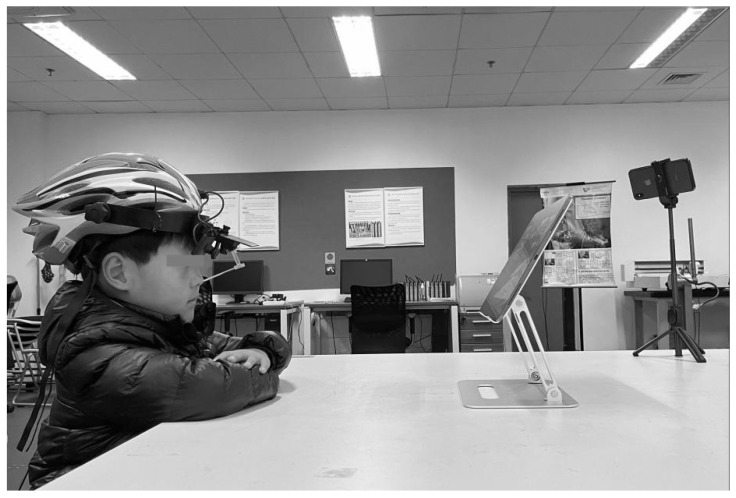
SMI headset eye tracker usage scenarios.

**Figure 3 healthcare-10-00621-f003:**
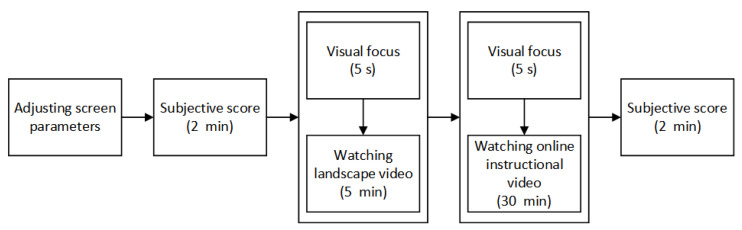
Experimental flow chart.

**Figure 4 healthcare-10-00621-f004:**
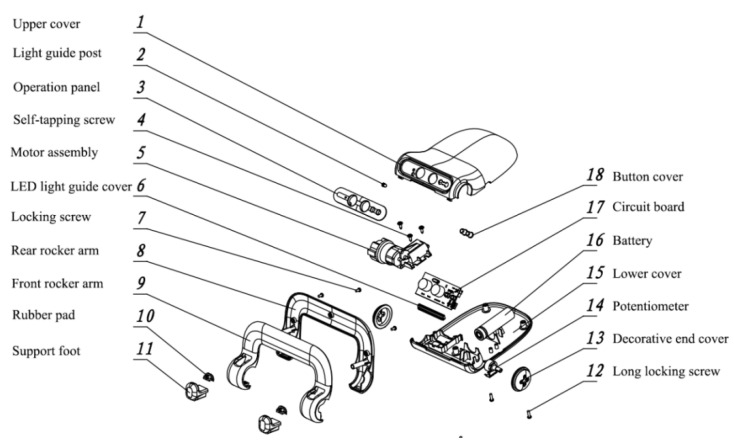
Structural explosion diagram of the self-accommodating vision protection device.

**Figure 5 healthcare-10-00621-f005:**
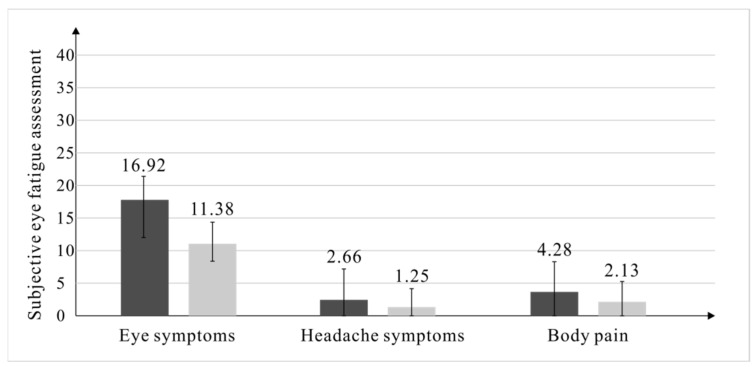
The post-test results of the subjective visual fatigue evaluation of the formal test and the verification experiment.

**Table 1 healthcare-10-00621-t001:** Paired *t*-test for the subjective visual fatigue assessment.

Subjective Evaluation	Pre-Test (M ± SD)	Post-Test (M ± SD)	*t*	*p*
Eye symptoms	0.12 ± 0.39	16.92 ± 1.37	−80.02	0.000 **
Headache symptoms	0.04 ± 0.03	2.66 ± 1.69	−11.05	0.000 **
Body pain	0.00 ± 0.00	4.28 ± 1.66	−18.03	0.000 **

Note: ** mean significant correlation at *p* < 0.05 and extremely significant correlation at *p* < 0.001, respectively, the same below.

**Table 2 healthcare-10-00621-t002:** Paired samples *t*-test for BF.

Time Periods	Blank Control Group (M ± SD)	Test Group (M ± SD)	*t*	*p*
0~5 min	7.91 ± 2.72	8.02 ± 2.54	−0.792	0.432
5~10 min	7.91 ± 2.72	7.85 ± 2.69	−0.403	0.688
10~15 min	7.91 ± 2.72	8.88 ± 2.78	−2.655	0.011 *
15~20 min	7.91 ± 2.72	9.69 ± 2.97	−3.720	0.001 **
20~25 min	7.91 ± 2.72	10.68 ± 3.07	−5.968	0.000 **
25~30 min	7.91 ± 2.72	9.48 ± 2.85	−3.627	0.001 **

Note: * and ** mean significant correlation at *p* < 0.05 and extremely significant correlation at *p* < 0.001, respectively, the same below.

**Table 3 healthcare-10-00621-t003:** Regression model output results.

Model	Correlation Coefficient	Coefficient of Determination	Coefficient of Adjust	Error of Standard Estimation
1	0.951 a	0.905	0.903	0.67229
2	0.968 b	0.937	0.935	0.55213
3	0.974 c	0.948	0.944	0.50963

Note: a is a predictor variable (constant): viewing angle; b is a predictor variable (constant): viewing angle, viewing distance; c is a predictor variable (constant): viewing angle, viewing distance, and apparent height.

**Table 4 healthcare-10-00621-t004:** Multivariate standard regression analysis of BF and different human–machine viewing parameters.

Model		Non-Standardized Regression Coefficient		Standard Regression Coefficient	*t*	*p*
		B	Standard Error			
1	Constant	−10.519	0.913		−11.516	0.000 **
	Viewing angle	0.891	0.042	0.951	21.410	0.000 **
2	Constant	−17.528	1.611		−10.879	0.000 **
	Viewing angle	0.598	0.069	0.639	8.709	0.000 **
	Viewing distance	0.068	0.014	0.361	4.916	0.000 **
3	Constant	−26.149	3.212		−8.140	0.000 **
	Viewing angle	0.474	0.075	0.507	6.294	0.000 **
	Viewing distance	0.054	0.013	0.290	4.052	0.000 **
	Visual height	0.026	0.009	0.221	3.028	0.004 *

Note: * and ** mean significant correlation at *p* < 0.05 and extremely significant correlation at *p* < 0.001, respectively, the same below.

## Data Availability

Not applicable.

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
