# Peer review of "A Study on the Design of Vision Protection Products Based on Children’s Visual Fatigue under Online Learning Scenarios"

_healthcare, 2022, doi:10.3390/healthcare10040621_

Round 1

Reviewer 1 Report

The manuscript titled "A Study on the Design of Vision Protection Products Based on Children’s Visual Fatigue under Online Learning Scenarios" by Di Feng et al. is an interesting paper and deserves publication after some minor English editing.

1- Figure 1: Please indicate on the figure the viewing angle alpha 1 and alpha 2.
2- Table 1: Please change Bubjective to Subjective.
3- Table 2 . Please correct the time periods for the blank control that is always 5 min. As it is the table is confusing.
4- Line 193-195: Please be clearer when describing the increase and decrease of the blinking frequency. The decrease is only at 30 minutes.
5- Line 228: Please change 0.280 to 0.290
6- Table 5: Please correct the time periods for the blank control that is always 5 min. As it is the table is a bit confusing.
7- Line 308-310: Please be clearer when describing the increase and decrease of the blinking frequency. The decrease is only at 30 minutes.

Author Response

Thank you for the valuable advice. We have adjusted the manuscript according to your suggestions.

  1. We modified the Figure 1 to indicate the viewing angle alpha 1 and alpha 2 by color contrast;
  2. We changed Bubjective to Subjective in Table;
  3. We changed the ordinate to 5 minutes because experimental data were obtained every 5 minutes;
  4. We added instructions for intercepting data within 30 minutes. “The date of 0-30 minutes are intercepted for display because it has been found that BF tends to be stable after 30min when children watch on the electronic screen.”
  5. We changed 0.280 to 0.290;
  6. We added a description of the horizontal and vertical coordinates and data sources in the figure 5. “Subjective eye fatigue assessment mainly included three aspects of eye symptoms, head symptoms and physical symptoms. Each question was divided into five grades from no to strong feeling, corresponding to 0-4 points respectively, with a total of 10 questions and a total of 40 points.” 
  7. We added instructions for intercepting data within 30 minutes. “The dates of 0-30 minutes are displayed because the data is basically flat after 30minutes.”

Reviewer 2 Report

Authors wrote an interesting article

the article is well written

I'd suggest to expand the limitations of the study

Author Response

Thank you for your positive comments. We added limitations of the study.

The added text is as follows:

“This experiment only studied three screen view parameters of children’s visual fatigue under online learning scenarios because of limited Laboratory equipment and time. Comparative experiments have not been recommended to determine the dominant influence factor.”

Reviewer 3 Report

This study that aims to investigate the effects of viewing angle, viewing height and viewing distance on visual fatigue when viewing electronic screens. This study is a nice contribution to the already existing literature.
Some minor comments:
-Line 60: please remove “to”
-Please provide details to the methods section: study start and end date, location where the study was conducted, Institutional review board approval.
-Line 85: Please move the following sentence to the beginning of the results section “In total, 52 grade 3-4 pupils (26 boys and 26 girls; mean age, 9.4 ± 0.6 years) participated in this study.”
-Table 1: please correct “bubjection evaluation”
-Line 179: please correct “pupils”
-Line 197: please correct “F”
-Line 199: please add “positive” correlation
-Line 228: please correct “0.280”
-Results are clear and well categorized.
-The discussion section discusses well the results from multiple angles and place them into context. Few suggestions: At the beginning of "Discussion", it is a good idea state in the first paragraph, the main findings of your work. After that, go ahead to discuss it. 
-The title is informative.

Author Response

Thank you for your positive comments. We have adjusted the manuscript according to your suggestions.

  1. 1. We removed “to”;
  2. We added study start and end date, location where the study was conducted, Institutional review board approval in Section 3.2;

The revised text is as follows:

“The study lasted from September 19, 2021 to January 15, 2022 in man-machine laboratory of Zhejiang University of Technology, which is approved by Institute of Industrial Design of Zhejiang University of Technology.”

  1. We changed Bubjective to Subjective in Table;
  2. We unified all “pupils” into “children” in the text;
  3. We modified “F” to “frequency”;
  4. We added “positive”;
  5. We changed 0.280 to 0.290;
  6. We placed the main findings of our work to the first paragraph of “Discussion”;

The revised text is as follows:

“This study is to investigate the effects of viewing angle, viewing height and viewing distance on visual fatigue when viewing electronic screens. Through experiments, it is found that there is a linear regression effect between human-machine viewing parameters and blink frequency, and the viewing angle, viewing height and viewing the effects of distance on visual fatigue were all positive effects.”